# Fault Prediction Based on Leakage Current in Contaminated Insulators Using Enhanced Time Series Forecasting Models

**DOI:** 10.3390/s22166121

**Published:** 2022-08-16

**Authors:** Nemesio Fava Sopelsa Neto, Stefano Frizzo Stefenon, Luiz Henrique Meyer, Raúl García Ovejero, Valderi Reis Quietinho Leithardt

**Affiliations:** 1Department of Electrical Engineering, Regional University of Blumenau, Rua São Paulo 3250, Blumenau 89030-000, Brazil; 2Fondazione Bruno Kessler, Via Sommarive 18, 38123 Trento, Italy; 3Department of Mathematics, Informatics and Physical Sciences, University of Udine, Via delle Scienze 206, 33100 Udine, Italy; 4Expert Systems and Applications Laboratory, E.T.S.I.I. of Béjar, Universidad de Salamanca, 37700 Salamanca, Spain; 5COPELABS, Lusófona University of Humanities and Technologies, Campo Grande 376, 1749-024 Lisboa, Portugal; 6VALORIZA, Research Center for Endogenous Resources Valorization, Instituto Politécnico de Portalegre, 7300-555 Portalegre, Portugal

**Keywords:** LSTM, GMDH, ANFIS, ensemble learning models, wavelet, time series forecasting

## Abstract

To improve the monitoring of the electrical power grid, it is necessary to evaluate the influence of contamination in relation to leakage current and its progression to a disruptive discharge. In this paper, insulators were tested in a saline chamber to simulate the increase of salt contamination on their surface. From the time series forecasting of the leakage current, it is possible to evaluate the development of the fault before a flashover occurs. In this paper, for a complete evaluation, the long short-term memory (LSTM), group method of data handling (GMDH), adaptive neuro-fuzzy inference system (ANFIS), bootstrap aggregation (bagging), sequential learning (boosting), random subspace, and stacked generalization (stacking) ensemble learning models are analyzed. From the results of the best structure of the models, the hyperparameters are evaluated and the wavelet transform is used to obtain an enhanced model. The contribution of this paper is related to the improvement of well-established models using the wavelet transform, thus obtaining hybrid models that can be used for several applications. The results showed that using the wavelet transform leads to an improvement in all the used models, especially the wavelet ANFIS model, which had a mean RMSE of 1.58 $\times \;10^{-3}$, being the model that had the best result. Furthermore, the results for the standard deviation were 2.18 $\times \;10^{-19}$, showing that the model is stable and robust for the application under study. Future work can be performed using other components of the distribution power grid susceptible to contamination because they are installed outdoors.

## 1. Introduction

Power distribution insulators are responsible for the electrical insulation and mechanical support of the cables of electrical power transmission and distribution aerial networks [1]. Because they are usually installed outdoors, they are constantly exposed to adverse weather conditions [2]. External agents must be considered when choosing the type of insulator to be used in a network because depending on the environment and the insulator chosen, they can compromise the proper operation of the grid and the life of the insulator itself. Typically, the insulators are manufactured using materials such as porcelain, glass, or polymers [3].

Contamination (pollution, salinity, biological agents) can make the surface of the insulator more conductive, increasing the possibility of partial discharges (PDs), dry band arcing, and consequently decreasing the efficiency of the system [4]. Contamination is a problem in places close to industries, agriculture, mining, coastal regions, or unpaved streets that may accumulate on the surface of the insulators [5]. Therefore, in some critical locations, cleaning or washing the insulators is required to mitigate insulation failure in the power system [6]. The layer of contamination deposited on the insulator when dry is not very conductive, but in the presence of moisture (light rain, fog), its conductivity can be increased by the formation of an electrolyte [7]. As the conductivity increases, PDs become more frequent and more intense, which can damage the surface of the insulator and evolve into dry band arcing that in turn may develop into a complete rupture known as flashover [8].

In the presence of a layer of wet contamination and leakage current on the surface of the insulator, heat dissipation, which can evaporate certain portions of this wet layer, drying out part of this conductive path, occurs [9]. This dry, non-conductive region is known as the dry band. These dry regions, which interrupt the conductive path due to evaporation, can concentrate an intense electric field, which can, in turn, lead to disruption of the air over the dry band, causing what is called dry band arcing and may damage the insulation [10]. One way to prevent a fault from becoming irreversible is to perform leakage current increase prediction through time series forecasting.

The major difficulty in using methods for time series forecasting is defining which model to use. Many authors support proposals for hybrid models in which several techniques are combined; however, a complete comparison to determine if the approach is the most adequate is usually not performed. In this paper, the models long short-term memory (LSTM) [11], group method of data handling (GMDH) [12], adaptive neuro fuzzy inference system (ANFIS) [13], bagging [14], boosting [15], random subspace [16], and stacking ensemble learning models [17] are compared. After defining the structure of each model that has the best performance for the used data set, the hyperparameters of the models and the use of wavelet transform, which has wide application for time series, are evaluated [18].

The remainder of this paper is structured as follows: Section 2 discusses the problem of contamination in insulators and the laboratory procedure used to perform the simulation of contamination accumulation. Section 3 presents the methods used for time series forecasting. Section 4 analyzes and discusses the results. Section 5 presents the conclusion.

## 2. Insulators Contamination

Since the power distribution insulators are usually installed outdoors, they are exposed to external agents such as sunlight, rain, and wind [19]. As time passes, it is natural for small amounts of particles such as dust or salt to be deposited on the insulator. This layer of dirt on the surface of the insulator is called contamination [20]. Contamination is one of the reasons of failure in power grid insulators [21]. This happens mainly in coastal areas due to the salinity present in the sea air, on unpaved streets, and in industrial regions with mining activities or chemical industries that generate suspended dirt. This contamination is distributed on the surface of the insulator in a non-uniform way [22].

The presence of contamination on the insulator’s surface does not mean that it needs to be replaced and normally has no harmful effect as long as moisture is not present [23]. However, in the presence of moisture, a conductive path, which decreases the insulation between the high-voltage phases and the ground, can be generated [24]. When cracks and fissures occur in the insulator, the rate of contamination deposition may accelerate and consequently increase its leakage current, making it more susceptible to partial discharge events and flashovers [25].

For polymeric insulators, Maraaba et al. [26] showed that insulators with up to 2 years of use have a better hydrophobic level compared to equivalent equipment with 15 years of operation. According to their study, there is a gradual reduction in hydrophobicity over time, which impairs the performance of the insulator. It is even recommended to replace the insulators after a field period longer than 19.2 years due to changes in their electrical and mechanical characteristics. The improvement of the network in the design phase has helped obtain more robust power systems [27,28].

In the presence of a contaminated and wet filament, dry band arcing occurs. This arcing dissipates energy through the joule effect, which can generate dry bands since the evaporation capacity is greater than the capacity to fill the affected region with water [29]. These dry bands bring the high voltage point closer to the earth, concentrating intense electric fields, which in turn can give rise to a series of dry band arcing [30]. The discharges can damage the surface of the polymeric insulator, giving rise to cracks that diminish its hydrophobic property and, in the long term, can lead to a complete rupture of the insulation [31].

Among the techniques for assessing contamination, the equivalent salt deposit density (ESDD) evaluates the amount of salt (NaCl) dissolved in a certain area measured in $mg/cm^{2}$. This value is defined by washing the insulator with a specific amount of water and then measuring the conductivity of the water [32]. From the ESDD measurement, it is possible to measure the current state of a specific insulator that can be extrapolated to several insulators in the same region with equivalent exposure time [33]. The major disadvantage of the ESDD method is the need to remove the insulator from the transmission line to perform an accurate measurement [34]. An alternative is to estimate the ESDD through the information of the leakage current [35], a consequence of the salt deposition, thus not requiring the removal of the insulator to perform the evaluation.

The accumulation of contamination and consequently PDs can cause the equipment to be at risk and lead to outages, which makes monitoring these insulators essential for electrical utility [36,37]. If maintenance is not performed and the insulation fails, a technician should perform corrective maintenance, searching and replacing the insulator in the field in an emergency manner [38].

One of the most effective way to assess surface insulation degradation is by monitoring the leakage current [39]. According to Ghunem et al. [40], the leakage current is the main cause of fires on poles, which may result in wildfires. When an insulator is contaminated, there may be an increase in leakage current until there is a disruptive failure [41]. The contamination accumulates over time and becomes embedded in the surface of the insulator [42]. The evaluation of the increase in leakage current can be an indication that a disruptive failure will occur [43].

There are a variety of techniques and equipment specialized in the detection of defective insulators. This analysis can be performed from visual inspection techniques [44] or even taking insulator samples for bench tests. The equipment commonly used for inspection of the network are ultrasound detectors [45], acoustic sensors [46], infrared [47], and ultra-violet cameras. Software with a focus on protection and security has been increasingly used [48], which can also be an alternative for use in monitoring the electrical power system. This maintenance is performed by field technicians who, when detecting possible defective insulators, clean or, if necessary, replace the insulator.

The use of machine learning to predict the increase in contamination levels, partial discharges, or/and faulty insulators has been growing recently. Because the failures do not follow a linear pattern, their monitoring is a challenging task. Models that use deep layers for time series prediction as well as models that combine simpler models to create a more robust structure are becoming popular [49]. Among the ensemble learning methods for time series forecasting, the highlights are bootstrap aggregation (bagging) [50], sequential learning (boosting) [51], random subspace [52], random forest [53], and stacked generalization [54].

### Laboratory Setup

To evaluate contamination in the laboratory, tests are performed in saline chambers, where controlled contamination situations are simulated on the surface of the insulator [55]. The experiments can be performed in two ways: The first method consists of using salty water to generate saline spray. In the second method, the contamination is applied directly to the surface of the insulator. In both methods, the salt levels on the surface of the insulator can be increased until the dielectric breakdown.

In the case of the experiment conducted in this paper with salt spray, the first method is applied. To simulate salt contamination that accumulates over time on the surface of the insulators, six insulators were mounted in a saline chamber. A 8.66 kV RMS 60 Hz was applied to the insulators (same phase), and the salt concentration was increased gradually. This voltage level is defined by the NBR 10621 (similar to IEC 60507) standard used by the electrical power utility. Specifically, this standard deals with the determination of the characteristics of supportability under artificial pollution for insulators in electric power grids for the 15 kV class. The arrangement of this experiment is shown in Figure 1.

Saline contamination was used in this paper because it is one of the contaminants that has the greatest impact on leakage current since salinity increases the surface conductivity of insulators, thus reducing their insulating capacity. With reduced insulation and increased leakage current, there is a greater chance of a flashover occurring [56]. Specifically, a saline spray chamber was used because it is an automated method of contamination that facilitates the evaluation of the experiment in relation to time, especially when it is necessary to carry out a prolonged experiment.

To monitor and record the applied voltage and resulting leakage current, an interface was developed in LabVIEW software. Each of the insulators was individually connected to the ground to measure the leakage current through a shunt resistor. Figure 2 presents the measured values of the leakage current during the experiment.

From the six analyzed insulators, two did not have flashover, and the respective leakage current values were measured until the end of the experiment. The other insulators were monitored only until the surface breakdown occurred. Using the signal recorded during the laboratory experiment, time series forecasting models were applied to evaluate the capability of predicting the development of a failure in relation to the increase in leakage current.

The discharges due to the increase in leakage current occur randomly. The purpose of the experiment was to subject the insulators to an extreme contamination condition, thus simulating insulators installed in the field and exposed to adverse conditions for several years. Since the discharge is random, there is no certainty that it will occur. For this reason, six insulators were used in an experiment under controlled conditions. The evolution of the leakage current in one of the insulators is enough to perform the time series prediction when there is a flashover. This means that at least one faulty insulator would be necessary to perform the evaluation. If there were no failures during the experiment, the experiment would have to be prolonged.

The experiment was conducted from 18 April 2022 to 29 April 2022, and more than 90,000 measurements were recorded. In this period, four insulators presented flashover, so the analysis could be performed in any of these components.

## 3. Time Series Forecasting

The values from the time series are used up to time *t*. This way, it is possible to predict the value in the future, t+P [57]. Thus, a mapping is created from the sample points *n*, sampled in each unit Δt in time,
(1)$$\hat{x}=x\left(t-\left(n-1\right)\Delta t\right),\ldots ,\;x\;\left(t-\Delta t\right),\;x\left(t\right),$$
to a predicted value,
(2)y(t+P)=f(x).

To predict the values of the next steps forward, the answers of the training sequence are changed on a time interval. Using the time series forecasting step ahead approach, each input sequence of the time step learns to predict the value of the next time step [58]. To obtain the expected values of future time steps, the output of the training sequences is shifted by a single time step [59]. From the time series evaluation, it is feasible to predict the development of flashover voltage, considering contamination conditions on electrical power insulators [60].

Several models can be used in time series forecasting, making the hoice of the appropriate model a difficult task. LSTM has been applied in deep learning by several authors due to its promising features in dealing with nonlinear data [61]. GMDH has performance advantages because it is an adaptive model that disregards neurons that do not help in the training process [62].

ANFIS has the advantages of fuzzy logic for time series forecasting [63]. The combination of simpler models makes the ensemble learning approach a promising alternative for forecasting [64], such as the bagging, boosting, random subspace, and stacking. These approaches will be explained and compared in this paper. The structure of these models is presented in Figure 3 and will be explained in this section.

### 3.1. LSTM

LSTM is a recurrent neural network used in deep learning that has become increasingly popular [65]. The major advantage of using the LSTM is that it can learn long-term dependencies, being able to handle nonlinear variations of the system, which is an important feature for time series forecasting [66].

An LSTM unit consists of a cell with an input port, an output port, and a forgetting port [67]. The unit remembers values at arbitrary time intervals and the three gates control the flow of information in and out of the cell [68]. The LSTM can be calculated according to the equations:(3)$$\begin{array}{c} i_{t}=\sigma_{g}\left(W_{i}x_{t}+R_{i}h_{t-1}+b_{i}\right),\\ f_{t}=\sigma_{g}\left(W_{f}x_{t}+R_{f}h_{t-1}+b_{f}\right),\\ o_{t}=\sigma_{g}\left(W_{o}x_{t}+R_{o}h_{t-1}+b_{o}\right), \end{array}$$
where $\sigma_{g}$ is the gate activation function, *W* and *R* are weight matrices, and *b* is the bias. These values are assigned to the network training [69].

In an LSTM recurrent unit, $h_{t-1}$ is the hidden state at the previous time step t−1 (short-term memory), $c_{t-1}$ is the cell state at previous time step t−1 (long-term memory), $x_{t}$ is the input vector at current time step *t*, $h_{t}$ is the hidden state at current time step *t*, and $c_{t}$ is the cell state at current time step *t* [70].

During the training phase, it is possible to define several optimizers, the most popular of them are stochastic gradient descent with momentum (SGDM) [71], adaptive moment estimation (ADAM) [72], and RMS propagation (RMSProp) [73]. In this paper, the preliminary evaluation will be performed using the SGDM with 50 hidden units, and after defining the best structure, all the mentioned optimizers will be evaluated.

### 3.2. GMDH

The group method of data handling is distinguished by being an inductive approach that performs the ranking of gradually complicated polynomial models and selects the best possible solution using an external criterion [74]. The external criterion is one major feature of GMDH, as it describes the requirements of the model. In this model, the number of hidden layers and the number of their neurons are determined automatically [75].

The GMDH selects the best structure that results in better performance to obtain an optimized network. When the minimum is no longer reduced with the previous layer, the network prediction error stops [76]. For a comparison of the models, a maximum of 50 neurons were initially used. The coefficients of GMDH are solved with regression methods for each pair of input variables $x_{i}$ and $x_{j}$, where:(4)$$\hat{y}\left(x_{1},\ldots ,\;x_{n}\right)=a_{0}+\sum_{i=1}^{n}a_{i}x_{i}+\sum_{i=1}^{n}\sum_{j=1}^{n}a_{ij}x_{i}x_{j}+\sum_{i=1}^{n}\sum_{j=1}^{n}\sum_{k=1}^{n}a_{ijk}x_{i}x_{j}x_{k}+\ldots$$
(5)$$G\left(x_{i},\;x_{j}\right)=a_{0}+a_{1}x_{i}+a_{2}x_{j}+a_{3}x_{i}^{2}+a_{4}x_{j}^{2}+ax_{i}x_{j}.$$

In this paper, the coefficients are estimated by the least-squares error (LSE) function:(6)$$\mathrm{LSE}=\left\{\begin{array}{c} \hat{y}\left(x_{1},\ldots ,\;x_{n}\right)=G\left(x_{i},\;x_{j}\right)\\ e=\sum_{n=1}^{N}{\left(y-\hat{y}\right)}^{2}\\ \frac{de}{da_{k}}=0,\;\;k=1,\;2,\;3,\;4,\;5. \end{array}\right.$$

To make the analysis easier, the results can be expressed in matrix form as:(7)$$a={\left(X^{T}X\right)}^{-1}X^{T}y$$
where
(8)$$X=\left\{\begin{array}{cccccc} 1 & x_{i1} & x_{j1} & x_{i1}x_{j1} & x_{i1}^{2} & x_{j1}^{2}\\ 1 & x_{i2} & x_{j2} & x_{i2}x_{j2} & x_{i2}^{2} & x_{j2}^{2}\\ 1 & x_{i3} & x_{j3} & x_{i1}x_{j2} & x_{i3}^{2} & x_{j3}^{2}\\ \vdots & \vdots & \vdots & \vdots & \vdots & \vdots \\ 1 & x_{in} & x_{jn} & x_{in}x_{jn} & x_{in}^{2} & x_{jn}^{2} \end{array}\right\}.$$

### 3.3. ANFIS

An ANFIS is a neural network based on the Takagi–Sugeno–Kang inference model. This method unites both the benefits of neural networks and fuzzy systems in the same structure. By using both characteristics of these methods, this approach can deal with systems involving imprecise and nonlinear data [77].

The behavior of ANFIS can be understood by observing variables related to membership functions, the relationship from inputs to outputs, and fuzzy rules. Given these features, the ANFIS model might be adopted for chaotic time series forecasting [78]. The model optimization was evaluated in two ways: being the backpropagation using a gradient descent to calculate all parameters and the hybrid method that performs a combination of backpropagation to calculate the input membership parameters and least-squares estimation to calculate the output membership parameters.

### 3.4. Ensemble Learning Models

Ensemble learning modeling is based on the divide-and-conquer challenge dedicated to enhancing the accuracy of models. Several weak learners individually perform a particular task, and when their results are combined, a more efficient model regarding the accuracy is achieved [79].

The ensemble learning approach has better results because every base model learns distinct features of the data, and then when the outputs are combined, the entire pattern of the data is learned by aggregating the weak learners (base models) [80]. Because of the suitability of the ensemble learning method to handle different types of data, applications can be found in various fields such as energy [81], security [82], public health [83], industry [84], and the environment [85].

The weak learners used in this paper are support vector regression (SVR). These base models were used, considering that they are an efficient approach for ensemble learning models [86]. The form of the SVR is defined by a convex optimization problem with linear constraints, denoted by:
(9a)$$\begin{array}{cc} \; & \min_{w,b,\sigma }\;\frac{1}{2}{\left||w|\right|}^{2}+C\sum_{i=1}^{m}\sigma_{i} \end{array}$$
(9b)$$\begin{array}{cc} \; & s.t.:\;y_{i}-\left(w\cdot x_{i}+b\right)\leq \varepsilon +\sigma_{i} \end{array}$$
(9c)$$\begin{array}{cc} \; & s.t.:\;\left(w\cdot x_{i}+b\right)-y_{i}\leq \varepsilon +\sigma_{i} \end{array}$$
(9d)$$\begin{array}{cc} \; & \sigma_{i}\geq 0,\forall i=1,\ldots ,m \end{array}$$
where w and *b* are the normal vector and the bias, respectively, of a training dataset ($x_{i}$, $y_{i}$), and ε is a margin of tolerance. The $\sigma_{i}$ is employed to transform (Equation 9) into a soft constraint (penalized by *C*), letting the optimization problem satisfy the constraint, even in ambiguous cases. To accomplish the relationship between input and output data,
(10)$$f\left(x\right)=w^{T}\phi \left(x\right)+b$$
where the forecasting values are f(x), and the mapping of the input vector x is ϕ; *b* and w are the coefficients calculated by the minimization of the risk function (*R*):(11)$$R\left(f\right)=\frac{1}{N}\sum_{i=1}^{N}L_{\varepsilon }\left(y_{i},w^{T}\phi \left(x_{i}\right)+b\right)$$
and the loss function ($L_{\varepsilon }$) is used to penalize the training errors, evaluated by:(12)$$L_{\varepsilon }\left(y,f\left(x\right)\right)=\left\{\begin{array}{cc} 0 & if\left|y-f\left(x\right)\right|\leq \varepsilon \\ \left|y-f\left(x\right)\right|-\varepsilon & otherwise. \end{array}\right.$$

Using the $L_{\varepsilon }$ in the regularized function, the task becomes a quadratic programming problem. The minimization of this function can be rewritten as an equivalent optimization problem, referred to as the *primal* problem:(13)$$\begin{array}{c} \min_{\left(w,b,\xi ,\xi^{*}\right)}\;R\left(w,b,\xi ,\xi^{*}\right)=\\ C\sum_{i=1}^{n}\left(\psi_{s}\left(\xi_{i}\right)+\psi_{s}\left(\xi_{i}^{*}\right)\right)+\frac{1}{2}{∥w∥}^{2} \end{array}$$
subject to
(14)$$\left\{\begin{array}{c} y_{i}-\langle w\cdot \phi (x_{i})\rangle -b\leq \left(1-\beta \right)\epsilon +\xi_{i};\\ \langle w\cdot \phi (x_{i})\rangle +b-y_{i}\leq \left(1-\beta \right)\epsilon +\xi_{i}^{*};\\ \xi_{i}^{*}\geq 0;\;\;\;\;\xi_{i}^{*}\geq 0\;\;\;\;\forall_{i} \end{array}\right.$$
where
(15)$$\psi_{s}\left(\xi \right)=\left\{\begin{array}{c} \frac{\xi^{2}}{4\beta \epsilon }\;\;\;\;if\;\;\;\;\xi \in \left[0,2\beta \epsilon \right];\\ \xi -\beta \epsilon \;\;\;\;if\;\;\;\;\xi \in [2\beta \epsilon ,+\infty ]. \end{array}\right.$$

Rewriting the *dual* problem:(16)$$\begin{array}{c} \min_{\left(\alpha ,\alpha^{*}\right)}\;\;R\left(\alpha ,\alpha^{*}\right)=\\ \frac{1}{2}\sum_{i=1}^{n}\sum_{i=1}^{n}\left(\alpha_{i}-\alpha_{i}^{*}\right)\left(\alpha_{j}-\alpha_{i}^{*}\right)K\left(x_{i},x_{j}\right)\\ -\sum_{i=1}^{n}\left(\alpha_{i}-\alpha_{i}^{*}\right)y_{i}+\sum_{i=1}^{n}\left(\alpha_{i}+\alpha_{i}^{*}\right)\left(1-\beta \right)\epsilon \\ +\frac{\beta \epsilon }{C}\sum_{i=1}^{n}\left(\alpha_{i}^{2}+\alpha_{i}^{*2}\right) \end{array}$$
subject to
(17)$$\begin{array}{c} 0\leq \alpha_{i}\leq C,\forall_{i},0\leq \alpha_{i}^{*}\leq C,\forall_{i}\\ and\\ \sum_{i=1}^{n}\left(\alpha_{i}-\alpha_{i}^{*}\right)=0. \end{array}$$

The Kernel functions (*K*) used in the SVR for this paper are linear (Equation 19), radial basis function (RBF) (Equation 18), and polynomial (Equation 20). For an overall evaluation, the quadratic programming (L1QP) [87], iterative single data algorithm optimization (ISDA) [88], and sequential minimal optimization (SMO) [89] optimizers were used. For the initial comparison of ensemble models, the linear Kernel function and the L1QP optimizer were adopted. After the best-fit model was defined, all the presented kernel functions and optimizers were evaluated.
(18)$$K\left(x_{j},x_{k}\right)=\exp\left(-{∥x_{j}-x_{k}∥}^{2}\right).$$
(19)$$K\left(x_{j},x_{k}\right)={x_{j}}^{\prime }\;x_{k}.$$
(20)$$K\left(x_{j},x_{k}\right)={\left(1+{x_{j}}^{\prime }\;x_{k}\right)}^{q}.$$

Several models are employed in the field of ensemble learning in which bootstrap aggregation (bagging) [90], sequential learning (boosting) [91], random subspace [92], and stacked generalization [93] can be highlighted. This grouping is intended to bring together the weakest multiple models to reduce their general susceptibility to the bias-variance, therefore making the prediction more robust.

#### 3.4.1. Bagging

The bagging ensemble method is a type of parallel method aimed to generate a more robust set of models than individual models that compose it (weak learners). Bagging is focused on reducing the variance of the resulting model. In this method, independent learners are considered independent of each other; then it is possible to train them simultaneously [94]. The bagging method can also be called bootstrap aggregation since the bootstrap sample is created initially for each model, and afterwards the model is aggregated, combined by the mean rule (in regression cases) [95].

#### 3.4.2. Boosting

The boosting ensemble learning is a process characterized as a sequential learning approach. Weak learners are not independently trained, focusing on the reduction of the bias of the individual models [96]. Indeed, for the regression tasks, the effectiveness of the boosting approach is due to the fact that after the result of the first weak model, the following models try to improve accuracy by fitting models to the residual of the previous models [97].

Using the regularization parameter, overfitting is avoided. In fact, the boosting paradigm trains new models iteratively, concentrating on observations that the prior models had more difficulty in predicting, which therefore makes the predictive model less impartial [98]. Since the goal is to reduce the bias of simpler predictors, it is proper to use a simpler model with higher bias and lower variance [99].

#### 3.4.3. Random Subspace

The random subspace ensemble learning model is a popular random sampling method that was introduced by Ho [100] to improve the performance of weak classifiers and to improve the classification accuracy of individual classifiers [101]. According to Pham et al. [102], random subspace is an ensemble approach where the original high-dimensional feature vector is randomly sampled to create the low-dimensional subspaces, and multiple classifiers are then combined on these random subspaces for the final decision.

#### 3.4.4. Stacked Generalization

Stacking ensemble learning combines several different predictive models into a single model working in layers or levels. This concept introduces meta-learning, which represents an asymptotically optimum learning system and is intended to minimize generalization errors by reducing the bias of its generalizers [103]. In fact, a stacked model is created from the predictions of weak learners, which are used as features. The characteristics allow the resulting model to cluster the initial models, so the model disregards the results that performed poorly [104].

### 3.5. Wavelet

For the purpose of comparing the proposed model, the signal will be filtered using the wavelet transform (WT) to assess whether the use of filters is promising for the application. In the WT, information is extracted from each signal segment and treated [105]. Initially, the wavelet energy coefficient is obtained after the signal decomposition by the wavelet packets transform (WPT), considering that the information on both sides of the spectrum is considered in this procedure [106].

The WPT performs a new decomposition in each interaction based on the coefficients of the previous iterations and thus indicates that the final number of coefficients depends on the number of iterations [107]. The orthogonal wavelet is decomposed into wavelet packages (WP); thus a vector tree structure is created. The structure is divided into two parts, the first being an approximation coefficient vector and the second a detailed result vector [108]. The WP function can be obtained by:(21)$$W_{j,\;\;k}^{n}\left(t\right)=2^{j/2}W^{n}\left(2^{j}t-k\right)$$
where *k* represents the translation operator, *j* is a scalable parameter, and *n* is the oscillation parameter. The first two WP functions for n=0 and n=1 are:(22)$$W_{0,0}^{0}\left(t\right)=\phi \left(t\right),\;W_{0,0}^{1}\left(t\right)=\psi \left(t\right).$$

Equation (Equation 21) represents the scale function, and Equation (Equation 22) represents the main function. The equations for n=2,3,…,N, can be defined according to the following relations:(23)$$W_{0,0}^{2n}\left(t\right)=\sqrt{2}\sum_{k}\delta \left(k\right)W_{1,k}^{n}\left(2t-k\right),$$
(24)$$W_{0,0}^{2n+1}\left(t\right)=\sqrt{2}\sum_{k}\zeta \left(k\right)W_{1,k}^{n}\left(2t-k\right),$$
where ζ(k) is a high-pass filter, and δ(k) is a low-pass filter [109]. The coefficients $\Omega_{j}^{n}\left(k\right)$ can be calculated by the product of x(t) by $W_{j,k}^{n}$, expressed by:(25)$$\Omega_{j}^{n}\left(k\right)=\int_{-\infty }^{\infty }x\left(t\right)W_{j,\;\;k}^{n}dt.$$

Each WP coefficient can be determined according to its frequency level. While the wavelet decomposes the elements of low frequency, the WPT decomposes the elements of all frequencies, so its use results in components of low and high frequencies [110]. Using the tree structure generated by the decomposition coefficients of approximation, an optimal binary value is obtained. The resulting subtree can be much smaller than the original; thus it can make the algorithm more efficient [111].

### 3.6. Considered Measures

Aiming to forecast the leakage current of the contaminated insulators, it is promising to evaluate the evolution of the failure through time series analysis, thus efficiently estimating the moment when the component is vulnerable to suffering a disruptive discharge [112].

The most commonly used measures of performance in relation to forecast error are root-mean-square error (RMSE), mean absolute percentage error (MAPE), and mean absolute error (MAE) [113], calculated as follows:(26)$$\mathrm{RMSE}=\sqrt{\frac{1}{n}\sum_{i=1}^{n}{\left(y_{i}-{\hat{y}}_{i}\right)}^{2}},$$
(27)$$\mathrm{MAPE}=\frac{1}{n}\sum_{i=1}^{n}\left|\frac{y_{i}-{\hat{y}}_{i}}{y_{i}}\right|,$$
(28)$$\mathrm{MAE}=\frac{1}{n}\sum_{i=1}^{n}\left|y_{i}-{\hat{y}}_{i}\right|,$$
where the error is calculated by the difference in the observed value $y_{i}$ to the predicted output ${\hat{y}}_{i}$ [114].

Typically for classification tasks, the metrics are based on the confusion matrix [115]; however, for prediction, the metrics are based on the error [116]. The coefficient of determination (R${}^{2}$) measures the adjustment of a statistical model to the observed values of a random variable. This is another widely used metric for evaluating regressions, calculated by:(29)$$R^{2}=1-\frac{RSS}{TTS},$$
where RSS is the relationship between the residual sum of squares, and TSS is the total sum of squares, given by:(30)$$RSS=\sum_{i=1}^{n}{\left(y_{i}-{\hat{y}}_{i}\right)}^{2},$$
(31)$$TSS=\sum_{i=1}^{n}{\left(y_{i}-{\bar{y}}_{i}\right)}^{2},$$
where ${\bar{y}}_{i}$ is the average of the observed value [117]. When the best model configurations were found, 100 runs were performed to evaluate the mean Equation (Equation 32), median Equation (Equation 33), standard deviation Equation (Equation 34), and variance Equation (Equation 35). The simulations were evaluated using an Intel Core I5-7400, 20 GB of random-access memory, with MATLAB software, version R2019.
(32)$${\bar{x}}_{i}=\frac{1}{m}\sum_{i=1}^{m}x_{i}.$$
(33)$$\mathrm{Median}=\left\{\begin{array}{c} x_{\left[m/2\right]},\;\mathrm{if}\;m\;\mathrm{is}\mathrm{even},\\ \frac{x_{\left[(m-1)/2\right]}+x_{\left[(m+1)/2\right]}}{2},\;\mathrm{if}\;m\;\mathrm{is}\mathrm{odd}. \end{array}\right.$$
(34)$$\mathrm{Std}\mathrm{Dev}.=\sqrt{\frac{1}{m-1}\sum_{i=1}^{m}{\left(x_{i}-{\bar{x}}_{i}\right)}^{2}}.$$
(35)$$\mathrm{Variance}=\frac{1}{m-1}\sum_{i=1}^{m}{\left(x_{i}-{\bar{x}}_{i}\right)}^{2},$$
where *m* is the number of performed runs (100 in this paper), $x_{i}$ is the result of the RMSE of each run (*i*), and ${\bar{x}}_{i}$ is the mean result of all the simulations.

## 4. Analysis of Results

The leakage current of the insulators was recorded with a time interval of one second between each record. During the experiment, 100,000 records were made, corresponding to approximately 27 h and 46 min of evaluation. As the time series is long-term, the down-sample method of order five is used to reduce the time series length. During the experiment, the water salinity went from 142.3 to 133,600.0 (μS), the pressure from 3 to 5 (BAR), and the water flow from 10.4 to 20.8 (mL/s); there was no significant change in temperature, humidity, and applied voltage.

The dielectric breakdown occurred in four out of the six tested insulators. Only the insulators that resulted in a disruptive discharge were considered for the analysis since this is the condition that should be predicted when an increase in insulator contamination occurs. One of the insulators at the end of the experimental analysis is shown in Figure 4. After recording the variation of leakage current over time due to the increased accumulation of contamination on the insulator surface, time series forecasting models were applied to evaluate the prediction capacity of the failure development. The best results for each structure are highlighted in bold in this section.

### 4.1. Time Series Forecasting Analysis

In this section, initially all models are evaluated with different structure configurations. After defining the best structure for the model, the hyperparameters of each model are evaluated. From the configuration definition that has better performance, a statistical analysis is conducted using the standard models. Then, based on the best model’s configuration, the depth of the wavelet transform is evaluated in each model. From the best use of wavelet, a final statistical analysis is performed to compare the results of the proposed hybrid models.

The comparison between the models regarding their structure in the LSTM and GMDH is related to the increase in the size of the neural network through the inclusion of more layers. The results are presented in Table 1.

All ensemble models had a higher computational effort than the other models until convergence, resulting in a considerably higher processing time to be computed. The GMDH was the fastest model for the required processing This occurred because it uses the number of layers and nodes according to the needs of the task, being an efficient adaptive model. Because of the greater number of maximum layers used, the GMDH needs more time to converge, similar to what occurs with the LSTM model when deeper layers are used. In this evaluation, the GMDH model was superior to the LSTM model in terms of considered error metrics, coefficient of determination, and time to convergence.

The coefficient of determination of the ensemble bagging, ANFIS using subtractive clustering, and GMDH with maximum use of two layers were higher than the other models and their different structures. In the LSTM, the use of a deeper network did not result in a progressive increase in model performance with respect to error evaluation. This shows that using a model based on deep learning may not always be the best alternative.

Regarding the MAPE and MAE, the model that had the lowest error result was the ANFIS subtractive clustering followed by the ensemble bagging and GMDH (max of two layers). The RMSE of these models was also lower compared to other structure configurations. These structures had the best results in this comparison. For this reason, their hyperparameters will be modified for a more complete evaluation of their capabilities. A comparison of the original (observed) signal and the predicted signal is presented in Figure 5.

### 4.2. Hyperparameter Optimization

To obtain models with better performance, the variation in the main configuration parameters of the structures that had superior results of each model was evaluated. Table 2 presents the results of the LSTM model using the SGDM, ADAM, and RMSprop optimizers, varying the number of hidden units.

When varying the hyperparameters of the LSTM model, there was no significant improvement, considering that the difference between the best and worst results were closer than the other compared models. Table 3 presents the results of the variation of the maximum number of neurons in the GMDH model. Specifically this parameter was evaluated because the model is adaptive, and it only needs to define the maximum number of layers and neurons.

There was not much variation in the results when the maximum number of neurons used by the GMDH was changed. The best result was obtained using the maximum of 80 neurons. This result shows that GMDH is promising for this application, given that even varying the hyperparameters of the model, the results remain with lower error compared to LSTM. The high value of the coefficient of determination makes it one of the fastest models to converge. The superior results of this model are due to the fact that it is an optimized model in which neurons that do not help in the learning phase are disregarded.

The next model in which the configuration of the hyperparameters was evaluated is the ANFIS model. Considering the subtractive clustering structure, the results of this evaluation regarding training form and influence radius are presented in Table 4.

There was a minor variation in the ANFIS subtractive clustering model by changing the influence radius using a hybrid optimization method. The lowest MAPE and MAE values occurred using the hybrid method with a radius of influence of 0.4. Using this method, the lowest RMSE value was achieved with a radius of influence of 0.2, ranking among the best coefficient of determination values. Considering that the MAPE and MAE using the classical backpropagation method were higher and the difference in RMSE and coefficient of determination were not high, the hybrid method proved to be more promising.

Table 5 shows the results of using the L1QP, ISDA, and SMO optimizers for the ensemble bagging model. For these optimizers, the linear, RBF, and polynomial Kernel functions are evaluated.

The ensemble bagging model took much longer to converge using the L1QP optimizer. This shows that an inadequate configuration can result in low performance and a high computational effort. The Kernel function that had the best coefficient of determination and lowest error results was the linear function, so the combination between the ISDA optimizer and the linear function had the best result in this analysis considering the metrics evaluated.

The best setup results in the hyperparameters evaluated in this section were: LSTM with 50 hidden units with one deeper layer using an SGDM optimizer, GMDH with a maximum of 2 layers and 80 neurons, ANFIS subtractive clustering with the hybrid method and influence radius of 0.2, and ensemble bagging with the ISDA optimizer with a linear Kernel function. These settings were used for the following analysis in this paper. The result of the statistical evaluation using these settings for the RMSE is presented in Table 6. In this evaluation, the ANFIS and ensemble models had better results with lower error, smaller variance, and standard deviation than GMDH and LSTM.

### 4.3. Application of Wavelet Transform

The results of the use of the wavelet transform on insulator 1 are presented in Figure 6. These results are presented in relation to a generic index, considering that after the use of the down-sample algorithm, the samples do not follow the time sequence of the experiment since the focus of the evaluation is the variation in the amplitude of the leakage current.

To evaluate the use of the wavelet transform, all models were tested using different combinations of depth for the wavelet packet tree. The results of this evaluation are presented in Table 7. The use of more than one node results in a loss of time series features, so this configuration was standardized.

The results of using the wavelet transform to reduce signal noise were promising in all models except for the LSTM in which the original signal prediction achieved better results of determination coefficients, time to convergence, and lower error. In GMDH, the use of the wavelet transform with two depth levels had better results than predicting the original signal. In the ANFIS model using two levels of depth, the RMS was close to the best result and had the best performance considering the other error metrics. For this reason, a depth of two levels was used in the ANFIS model, as in GMDH.

The ensemble model had a promising result using three levels in the wavelet transform. Only MAE and MAPE had superior results on the original signal after using the down-sample algorithm. The use of the wavelet transform had promising results when it was applied after the down-sample algorithm. When the analysis was performed using the down-sample before the wavelet transform, all models had inferior results, not being a suitable strategy for this analysis.

Considering the use of the best configuration of the wavelet transform, 100 runs were performed, and the statistical results regarding RMSE are presented in Table 8. As can be observed, the models that had the best performance are the ANFIS model and the ensemble, which were the same models that had the best performance without the use of the wavelet transform.

An interesting result is that the LSTM had the worst performance in both analyses, being a model that is not suitable for this evaluation. Many authors have used the LSTM; however, this model may not be the most appropriate for forecasting depending on the signal used, as presented in this paper.

## 5. Conclusions

The use of time series prediction models to evaluate the development of faults in insulators based on leakage current shows promise. Several models can be successfully applied to accomplish this task. The increase in contamination results in a consequent increase in leakage current until an electrical discharge occurs, which is an adequate way of assessing the development of the adverse condition to result in a failure. For this reason, leakage current must be monitored to keep the electrical power system operational.

The ANFIS subtractive clustering and the ensemble bagging stood out in predicting the leakage current, considering that they had lower error and better coefficient of determination. The results show that the structure of the model has a major influence on its performance. Then it is necessary to perform a comparative analysis of all variations of the model to have an optimized algorithm. The statistical results showed that ANFIS is a stable model, resulting in low variance when several simulations are performed.

The application of the wavelet transform resulted in an improvement in the predictive ability of the evaluated models, being a promising technique for noise reduction without the loss of signal characteristics. With a mean RMSE of 1.58 $\times \;10^{-3}$, the wavelet ANFIS had the best results with lower error and variance; comparatively, this model had a 53.74% lower error (RMSE) than the wavelet ensemble which was the second best model in this study. Most of the models presented stability when several simulations were performed, showing that these methods are reliable for the application presented in this paper. In particular, the wavelet ANFIS had a variance result of 2.18 $\times \;10^{-19}$, which is considerably lower than all the other models, thus proving to be the most stable model.

Based on the results, future work can be performed to develop an embedded system for monitoring leakage current and indicating vulnerability to a disruptive discharge in distribution insulators. The leakage current proves to be a suitable indicator for monitoring the conditions of the power electrical system; this measurement can be applied to other insulating components of the electrical power grid.

## Figures and Tables

**Figure 1 sensors-22-06121-f001:**
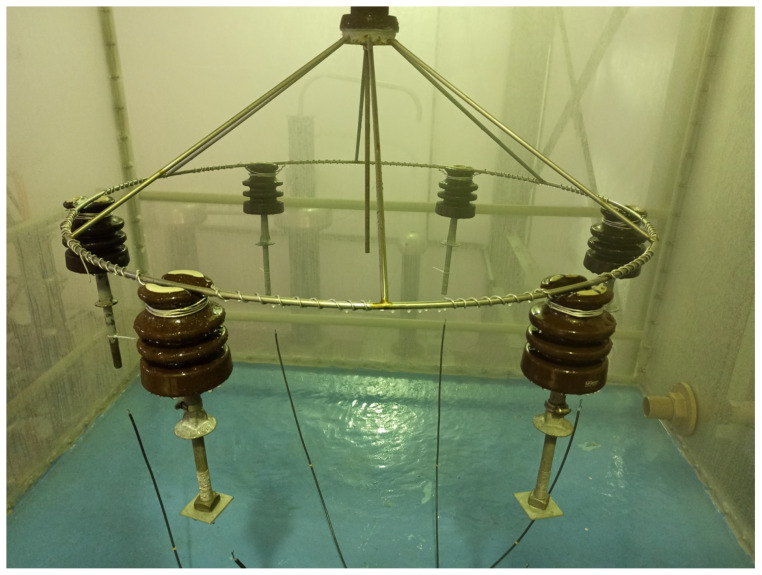
Insulators in a saline chamber experiment.

**Figure 2 sensors-22-06121-f002:**
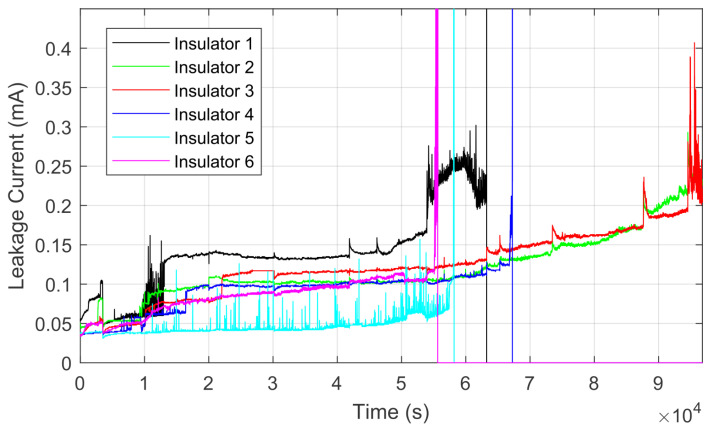
Leakage current of the contaminated insulators.

**Figure 3 sensors-22-06121-f003:**
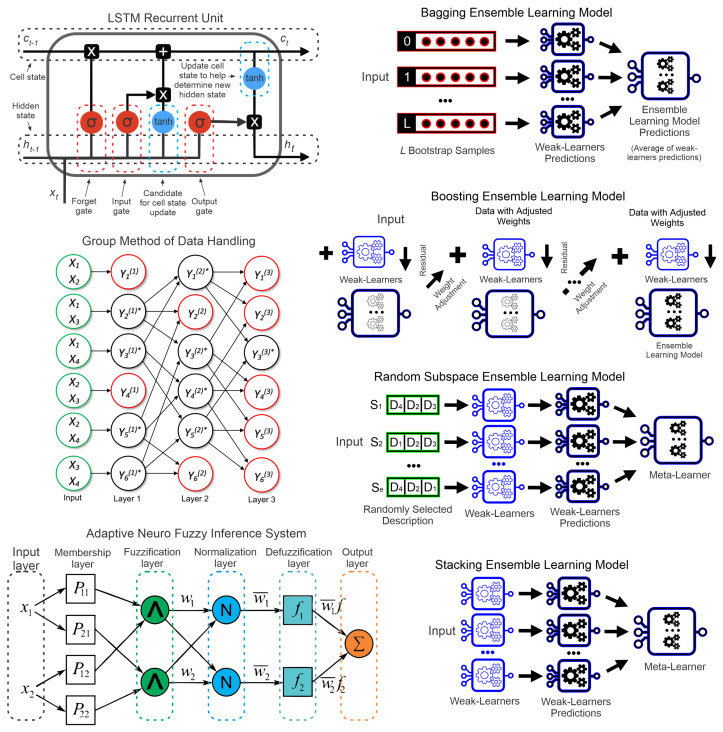
Structure of the considered models.

**Figure 4 sensors-22-06121-f004:**
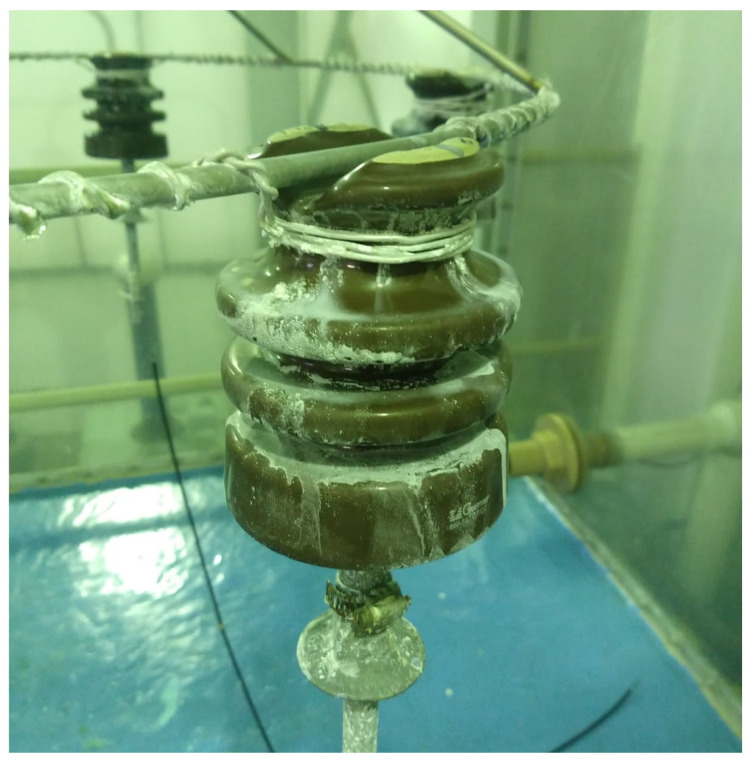
Insulator with salt contamination accumulated on its surface at the end of the experiment.

**Figure 5 sensors-22-06121-f005:**
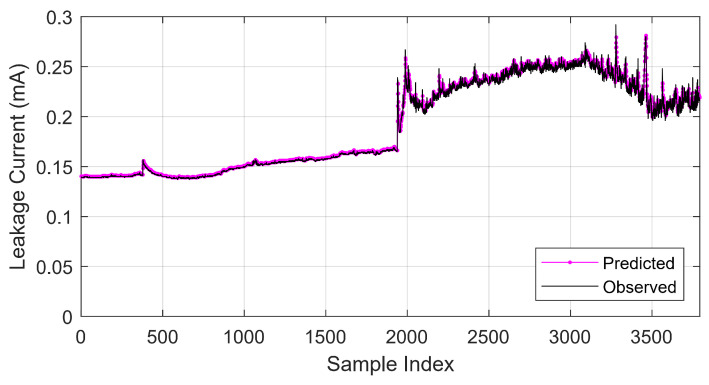
Predicted signal compared to observed signal.

**Figure 6 sensors-22-06121-f006:**
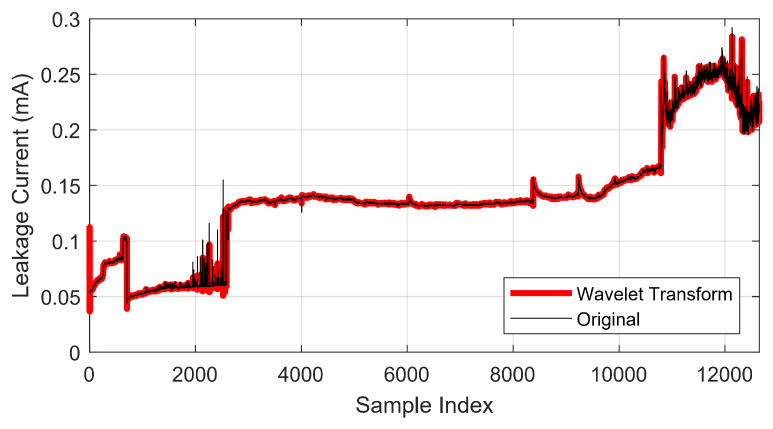
Wavelet transform evaluation.

**Table 1 sensors-22-06121-t001:** Overall comparison of the models.

Model	Structure	RMSE	MAPE	MAE	R${}^{2}$	Time (s)
LSTM	1 Deeper Layer	$2.35\times 10^{-2}$	**6.60 $\times \;10^{-2}$**	**1.56 $\times \;10^{-3}$**	**0.7104**	**75.86**
	2 Deeper Layers	3.01 $\times \;10^{-2}$	8.07 $\times \;10^{-2}$	1.95 $\times \;10^{-2}$	0.5269	107.22
	3 Deeper Layers	4.49 $\times \;10^{-2}$	1.37 $\times \;10^{-1}$	3.17 $\times \;10^{-2}$	0.0506	155.01
	4 Deeper Layers	5.22 $\times \;10^{-2}$	1.63 $\times \;10^{-1}$	3.75 $\times \;10^{-2}$	0.4208	182.07
GMDH	1 Max. Layer	4.79 $\times \;10^{-3}$	3.32 $\times \;10^{-3}$	7.18 $\times \;10^{-4}$	0.9880	2.21
	2 Max. Layers	**4.35 $\times \;10^{-3}$**	**1.90 $\times \;10^{-3}$**	**4.70 $\times \;10^{-4}$**	**0.9901**	2.90
	3 Max. Layers	5.10 $\times \;10^{-3}$	9.06 $\times \;10^{-3}$	2.10 $\times \;10^{-3}$	0.9864	**1.92**
	4 Max. Layers	6.00 $\times \;10^{-2}$	3.11 $\times \;10^{-2}$	7.30 $\times \;10^{-3}$	0.8819	4.68
ANFIS	FCM	1.15 $\times \;10^{-2}$	2.38 $\times \;10^{-2}$	4.85 $\times \;10^{-3}$	0.9304	**25.08**
	Grid Partitioning	5.23 $\times \;10^{-3}$	5.68 $\times \;10^{-3}$	1.45 $\times \;10^{-3}$	0.9857	92.53
	Subt. Clustering	**4.48 $\times \;10^{-3}$**	**3.29 $\times \;10^{-5}$**	**3.43 $\times \;10^{-5}$**	**0.9895**	73.75
Ensemble	Bagging	**4.19 $\times \;10^{-3}$**	**1.88 $\times \;10^{-3}$**	**3.16 $\times \;10^{-4}$**	**0.9909**	4443.30
	Boosting	3.40 $\times \;10^{-2}$	1.27 $\times \;10^{-1}$	2.73 $\times \;10^{-2}$	0.3971	31,874.70
	Random Subsp.	4.94 $\times \;10^{-3}$	1.09 $\times \;10^{-2}$	2.21 $\times \;10^{-3}$	0.9872	25,155.34
	Stacking	5.04 $\times \;10^{-2}$	1.56 $\times \;10^{-1}$	3.59 $\times \;10^{-2}$	0.3255	**1295.48**

**Table 2 sensors-22-06121-t002:** Evaluation of LSTM hyperparameters.

Optimizer	Hidden Units	RMSE	MAPE	MAE	R${}^{2}$	Time (s)
SGDM	10	5.40 $\times \;10^{-2}$	1.67 $\times \;10^{-1}$	3.85 $\times \;10^{-2}$	0.5226	81.53
	20	3.73 $\times \;10^{-2}$	1.12 $\times \;10^{-1}$	2.60 $\times \;10^{-2}$	0.2737	78.02
	30	2.32 $\times \;10^{-2}$	7.09 $\times \;10^{-2}$	1.63 $\times \;10^{-2}$	0.7180	71.09
	40	2.77 $\times \;10^{-2}$	8.48 $\times \;10^{-2}$	1.95 $\times \;10^{-2}$	0.5981	**70.62**
	50	**1.78 $\times \;10^{-2}$**	**4.82 $\times \;10^{-2}$**	**1.15 $\times \;10^{-2}$**	**0.8339**	74.71
ADAM	10	5.61 $\times \;10^{-2}$	1.79 $\times \;10^{-1}$	4.08 $\times \;10^{-2}$	0.6419	**71.29**
	20	5.64 $\times \;10^{-2}$	1.76 $\times \;10^{-1}$	4.05 $\times \;10^{-2}$	0.6601	72.94
	30	3.33 $\times \;10^{-2}$	1.02 $\times \;10^{-1}$	2.34 $\times \;10^{-2}$	0.4206	74.13
	40	2.97 $\times \;10^{-2}$	8.85 $\times \;10^{-2}$	2.06 $\times \;10^{-2}$	0.5396	74.16
	50	**1.82 $\times \;10^{-2}$**	**4.84 $\times \;10^{-2}$**	**1.16 $\times \;10^{-2}$**	**0.8274**	74.43
RMSprop	10	5.13 $\times \;10^{-2}$	1.68 $\times \;10^{-1}$	3.79 $\times \;10^{-2}$	0.3732	**72.47**
	20	5.13 $\times \;10^{-2}$	1.75 $\times \;10^{-1}$	3.90 $\times \;10^{-2}$	0.3762	73.32
	30	4.50 $\times \;10^{-2}$	1.51 $\times \;10^{-1}$	3.38 $\times \;10^{-2}$	0.0566	73.74
	40	**3.10 $\times \;10^{-2}$**	**9.16 $\times \;10^{-2}$**	**2.14 $\times \;10^{-2}$**	**0.4995**	75.56
	50	3.73 $\times \;10^{-2}$	1.33 $\times \;10^{-1}$	2.92 $\times \;10^{-2}$	0.2722	73.08

**Table 3 sensors-22-06121-t003:** Evaluation of GMDH hyperparameters.

Max Neurons	RMSE	MAPE	MAE	R${}^{2}$	Time (s)
10	4.15 $\times \;10^{-3}$	7.79 $\times \;10^{-4}$	2.04 $\times \;10^{-4}$	0.9910	0.26
20	4.58 $\times \;10^{-3}$	5.61 $\times \;10^{-3}$	1.31 $\times \;10^{-3}$	0.9891	0.34
30	4.49 $\times \;10^{-3}$	1.41 $\times \;10^{-3}$	2.92 $\times \;10^{-4}$	0.9903	0.25
40	4.59 $\times \;10^{-3}$	5.89 $\times \;10^{-3}$	1.30 $\times \;10^{-3}$	0.9890	0.24
50	4.30 $\times \;10^{-3}$	2.57 $\times \;10^{-3}$	6.23 $\times \;10^{-4}$	0.9903	0.24
60	4.22 $\times \;10^{-3}$	4.44 $\times \;10^{-4}$	1.28 $\times \;10^{-4}$	0.9907	0.24
70	4.42 $\times \;10^{-3}$	3.43 $\times \;10^{-3}$	8.15 $\times \;10^{-4}$	0.9905	0.26
80	**4.09 $\times \;10^{-3}$**	**3.75 $\times \;10^{-4}$**	**1.22 $\times \;10^{-4}$**	**0.9912**	0.26
90	4.45 $\times \;10^{-3}$	3.70 $\times \;10^{-3}$	8.18 $\times \;10^{-4}$	0.9897	0.24
100	4.34 $\times \;10^{-3}$	1.08 $\times \;10^{-3}$	2.80 $\times \;10^{-4}$	0.9902	**0.23**

**Table 4 sensors-22-06121-t004:** Evaluation of ANFIS subtractive clustering hyperparameters.

Method	Radius	RMSE	MAPE	MAE	R${}^{2}$	Time (s)
Hybrid	0.2	**4.15 $\times \;10^{-3}$**	5.92 $\times \;10^{-4}$	9.44 $\times \;10^{-5}$	**0.9910**	28.34
	0.4	4.48 $\times \;10^{-3}$	**3.29 $\times \;10^{-5}$**	**3.43 $\times \;10^{-5}$**	0.9895	28.81
	0.6	4.16 $\times \;10^{-3}$	4.86 $\times \;10^{-4}$	7.06 $\times \;10^{-5}$	**0.9910**	29.39
	0.8	4.16 $\times \;10^{-3}$	1.25 $\times \;10^{-3}$	2.39 $\times \;10^{-4}$	**0.9910**	**25.68**
	1.0	4.17 $\times \;10^{-3}$	6.83 $\times \;10^{-4}$	1.14 $\times \;10^{-4}$	0.9909	25.74
Backpropag.	0.2	7.60 $\times \;10^{-3}$	3.57 $\times \;10^{-2}$	6.41 $\times \;10^{-3}$	0.9698	25.18
	0.4	4.09 $\times \;10^{-3}$	2.00 $\times \;10^{-3}$	3.98 $\times \;10^{-4}$	0.9913	**24.96**
	0.6	**4.06 $\times \;10^{-3}$**	**1.74 $\times \;10^{-3}$**	**3.02 $\times \;10^{-4}$**	**0.9914**	25.12
	0.8	7.66 $\times \;10^{-3}$	3.58 $\times \;10^{-2}$	6.52 $\times \;10^{-3}$	0.9693	30.52
	1.0	8.17 $\times \;10^{-3}$	3.89 $\times \;10^{-2}$	7.12 $\times \;10^{-3}$	0.9651	25.51

**Table 5 sensors-22-06121-t005:** Evaluation of ensemble bagging hyperparameters.

Optimizer	Kernel	RMSE	MAPE	MAE	R${}^{2}$	Time (s)
L1QP	Linear	**4.19 $\times \;10^{-3}$**	**1.88 $\times \;10^{-3}$**	**3.16 $\times \;10^{-4}$**	**0.9909**	**4443.30**
	RBF	1.05 $\times \;10^{-1}$	3.42 $\times \;10^{-1}$	7.76 $\times \;10^{-2}$	0.7926	5256.65
	Polynomial	2.98 $\times \;10^{-2}$	1.43 $\times \;10^{-2}$	3.18 $\times \;10^{-3}$	0.5363	5763.47
ISDA	Linear	**4.17 $\times \;10^{-3}$**	**1.27 $\times \;10^{-3}$**	**1.98 $\times \;10^{-4}$**	**0.9909**	25.33
	RBF	9.70 $\times \;10^{-2}$	3.14 $\times \;10^{-1}$	7.14 $\times \;10^{-2}$	0.9143	**11.52**
	Polynomial	9.68 $\times \;10^{-2}$	3.14 $\times \;10^{-1}$	7.13 $\times \;10^{-2}$	0.8935	11.83
SMO	Linear	**4.23 $\times \;10^{-3}$**	**3.19 $\times \;10^{-3}$**	**5.74 $\times \;10^{-4}$**	**0.9906**	8.09
	RBF	1.06 $\times \;10^{-1}$	3.44 $\times \;10^{-1}$	7.82 $\times \;10^{-2}$	0.8680	**6.06**
	Polynomial	1.06 $\times \;10^{-1}$	3.44 $\times \;10^{-1}$	7.80 $\times \;10^{-2}$	0.8132	6.67

**Table 6 sensors-22-06121-t006:** Statistical assessment.

Model	Mean	Median	Std. Dev.	Variance
LSTM	2.17 $\times \;10^{-2}$	2.11 $\times \;10^{-2}$	3.30 $\times \;10^{-3}$	1.09 $\times \;10^{-5}$
GMDH	4.45 $\times \;10^{-3}$	4.38 $\times \;10^{-3}$	3.02 $\times \;10^{-3}$	9.12 $\times \;10^{-6}$
ANFIS	**4.15 $\times \;10^{-3}$**	**4.15 $\times \;10^{-3}$**	**8.72 $\times \;10^{-18}$**	**7.60 $\times \;10^{-35}$**
Ensemble	4.20 $\times \;10^{-3}$	4.19 $\times \;10^{-3}$	5.77 $\times \;10^{-5}$	3.33 $\times \;10^{-9}$

**Table 7 sensors-22-06121-t007:** Analysis using the wavelet transform.

Model	Depth	RMSE	MAPE	MAE	R${}^{2}$	Time (s)
Wavelet LSTM	1	3.83 $\times \;10^{-2}$	1.18 $\times \;10^{-1}$	2.72 $\times \;10^{-2}$	0.2314	86.24
	2	3.72 $\times \;10^{-2}$	1.13 $\times \;10^{-1}$	2.61 $\times \;10^{-2}$	0.2776	**70.33**
	3	**3.28 $\times \;10^{-2}$**	**1.12 $\times \;10^{-1}$**	**2.48 $\times \;10^{-2}$**	**0.4376**	73.87
	4	4.01 $\times \;10^{-2}$	1.46 $\times \;10^{-1}$	3.17 $\times \;10^{-2}$	0.1595	77.55
Wavelet GMDH	1	3.98 $\times \;10^{-3}$	5.79 $\times \;10^{-3}$	1.35 $\times \;10^{-3}$	0.9917	**0.22**
	2	**3.01 $\times \;10^{-3}$**	**4.09 $\times \;10^{-3}$**	**9.50 $\times \;10^{-4}$**	**0.9953**	0.29
	3	5.81 $\times \;10^{-3}$	1.01 $\times \;10^{-2}$	2.30 $\times \;10^{-3}$	0.9824	0.23
	4	3.94 $\times \;10^{-3}$	6.11 $\times \;10^{-3}$	1.42 $\times \;10^{-3}$	0.9919	0.24
Wavelet ANFIS	1	**1.56 $\times \;10^{-3}$**	5.31 $\times \;10^{-4}$	1.17 $\times \;10^{-4}$	**0.9987**	35.93
	2	1.58 $\times \;10^{-3}$	**5.48 $\times \;10^{-5}$**	**1.28 $\times \;10^{-5}$**	**0.9987**	42.54
	3	1.57 $\times \;10^{-3}$	1.76 $\times \;10^{-4}$	3.90 $\times \;10^{-5}$	**0.9987**	33.86
	4	1.57 $\times \;10^{-3}$	2.03 $\times \;10^{-4}$	4.52 $\times \;10^{-5}$	**0.9987**	**33.05**
Wavelet Ensemble	1	2.64 $\times \;10^{-3}$	8.46 $\times \;10^{-3}$	1.71 $\times \;10^{-3}$	0.9964	21.03
	2	3.62 $\times \;10^{-3}$	1.44 $\times \;10^{-2}$	2.88 $\times \;10^{-3}$	0.9931	**17.65**
	3	**2.46 $\times \;10^{-3}$**	**7.01 $\times \;10^{-3}$**	**1.43 $\times \;10^{-3}$**	**0.9968**	20.08
	4	3.12 $\times \;10^{-3}$	1.15 $\times \;10^{-2}$	2.31 $\times \;10^{-3}$	0.9949	19.90

**Table 8 sensors-22-06121-t008:** Statistical evaluation of models with wavelet transform.

Model	Mean	Median	Std. Dev.	Variance
Wavelet LSTM	2.08 $\times \;10^{-2}$	2.06 $\times \;10^{-2}$	3.48 $\times \;10^{-3}$	1.21 $\times \;10^{-5}$
Wavelet GMDH	4.39 $\times \;10^{-3}$	4.29 $\times \;10^{-3}$	1.45 $\times \;10^{-3}$	2.10 $\times \;10^{-6}$
Wavelet ANFIS	**1.58** $\times \;10^{-3}$	**1.58 $\times \;10^{-3}$**	**2.18 $\times \;10^{-19}$**	**4.75 $\times \;10^{-38}$**
Wavelet Ensemble	2.94 $\times \;10^{-3}$	2.91 $\times \;10^{-3}$	3.12 $\times \;10^{-4}$	9.76 $\times \;10^{-8}$

## Data Availability

For future comparisons, the recorded data are available at https://github.com/SFStefenon/LeakageCurrent (accessed on 1 August 2022).

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
