# Peer review of "Fault Prediction Based on Leakage Current in Contaminated Insulators Using Enhanced Time Series Forecasting Models"

_sensors, 2022, doi:10.3390/s22166121_

Round 1
Reviewer 1 Report
1.The abstract focus on introduces the background of the study subject and there is only one sentence talking about the contribution of this manuscript. Please modify the abstract, it’s need contain the background of the study subject, the contribution of this manuscript, verification results. 2.This manuscript uses a lot of space to introduces the exciting models, which can simplify quoted from references. I suggest the author to fuse the first and second sections together. 3.The innovation of this paper is not clear, the author should prove some attractive conclusions to readers instead of summarizing the existing model algorithm. I suggest the author could expand the context of the paper. 4.There is some confusion in the simulation verification, which is not easy for readers to understand. The description is not sufficientAuthor Response
Please find the Response to Reviewers attached.
Thank you.

Reviewer 2 Report
In this paper, the authors investigate the use of time-series prediction methods to address the failure of contaminated power distribution insulators formed on the surface due to exposure to external media such as sunlight, rain, and wind. The author studied the insulator in the accumulation of salt pollution with time, the relationship between the applied voltage and the leakage current, and the statistical results show that ANFIS is a stable model in several existing research models by monitoring the leakage current to maintain the operation of the power system. The most interesting point is that the authors use the time-series prediction model to evaluate the insulator failure based on the leakage current, and compare the model results in several available models. The results may provide a potential approach to troubleshooting insulator faults through multiple models combined with the detection of leakage currents, with a unique advantage for keeping the power system running. This study is interesting and worthwhile, and I recommend publication after addressing the following some questions.
1.Correct missing punctuation marks. Such as the comma is missing after the grid in line 1, and the comma should be added at best after the fog) in line 31.
2.Why did the authors use the salt spray method when simulating contamination? What are the advantages of the salt spray?
3.Why is there no control group in the experiment in the saline chamber? Are there graphic changes similar to figure 2 with the passage of time by using distilled water spray?
4.In this paper, you have only compared the advantages and disadvantages of each model, and you cannot determine which model is best suited for insulator failure prediction. Please make some explanation.
5.In the experiment, the flashover have not been found for two insulators, and the author did not give an explanation. Is it the accident of the experiment or the insulator itself? The authors should consider if there is an effect on the results of the next experiment.
Author Response
Please find the Response to Reviewers attached.
Thank you.

Reviewer 3 Report
The article "Leakage Current Based Fault Prediction in Insulator Insulators Using Enhanced Time Series Forecasting Models" is interesting and is within the scope of the Sensors journal. Strategies to minimize the influence of contamination in relation to leakage current and its progression towards a disruptive discharge is a very interesting topic from a performance enhancement point of view.
I would recommend revising the abstract to include in it same of the most important numerical results obtained with the different tested models, in order to highlight much more the results obtained. It would be interesting indicating which one of the models performs best for the purposes stated in the article.
As in the abstract, the conclusions should clearly indicate which of all the methods analyzed is the best suited to the objectives set.
Author Response

(The authors gave the same response as above.)
